# Towards Understanding Primacy and Recency Effects in Mamba: A Mechanistic Perspective

## Abstract

We uncover a sparse subset of channels in Mamba's selective state-space block that serves as a substrate for early-input retention. Identified through structured recall tasks, ablating these channels selectively degrades early positional recall. Input periodicity systematically shifts Mamba's discretization gate, amplifying the "lost-in-the-middle" effect by reallocating information across positions. Primacy and periodicity-driven effects, combined with recency, yield the characteristic U-shaped recall curve, aligning with effects known in Transformers but underexplored in state-space models. We further examine how distractor tokens affect Mamba's temporal dynamics: recency, sustained by an exponential-decay mechanism, collapses under distraction as it moves the queried items deeper in the sequence. Finally, we demonstrate that the same sparse subset of channels transfers beyond recall. Intervening on them degrades the performance on downstream long-context understanding tasks, indicating that they function as data-agnostic long-term memory carriers. These results provide a common mechanistic picture of Mamba's temporal profile, linking primacy, recency, and input periodicity.

## 1 Introduction

Primacy and recency, the tendencies to recall items presented at the beginning and end of a sequence more accurately than those in the middle, are long-established findings in cognitive science (Ebbinghaus, 1913; Atkinson & Shiffrin, 1968). In humans, the cause of primacy is hypothesized to be the better encoding of initial items due to greater rehearsal opportunity, while recency can be explained by short-term memory, where new learning overwrites the old learning (Greene et al., 2000).

These position-based recall phenomena have been well studied in the context of the traditional transformer architecture. Janik (2023) showed empirically that tokens at the beginning and end of a sequence are recalled more reliably than those in the middle. According to Wu et al. (2025), this asymmetry can be understood as a consequence of mechanistic features of the transformer design. In particular, causal masking draws attention toward earlier tokens, while relative positional encodings such as RoPE (Su et al., 2024) emphasize recent ones.

Recently, state-space language models (SSMs) have begun to garner significant research interest, due to their ability to scale more efficiently with sequence length compared to transformers, and offering efficient memory complexity for long-context inputs (Gu & Dao, 2023; Lv et al., 2025). Yet despite this progress, evidence of position-dependent effects in this family is still fragmentary. Primacy has been documented in the Structured State Space sequence model (S4) (Gu et al., 2021; Morita, 2025), while recency has been studied in Mamba, specifically linked to exponential decay dynamics (Wang et al., 2025). These observations suggest that state-space models, despite relying on recurrent updates rather than attention, are also subject to systematic primacy and recency biases, yet a room remains in the research literature for a holistic examination of effects related to these phenomena.

Importantly, explanations for primacy in transformers, where early-token recall improves with depth due to causal masking and exponential propagation of information (Wu et al., 2025), are not exclusive to attention-based models. SSMs also impose a causal structure through recurrent updates, with gates applied both to inputs (input-dependent decay) and to states (state-dependent decay). This makes them natural candidates for similar early-information advantages. Yet in Mamba (Gu & Dao, 2023), while recency has been linked to exponential decay, a principled account of primacy and its

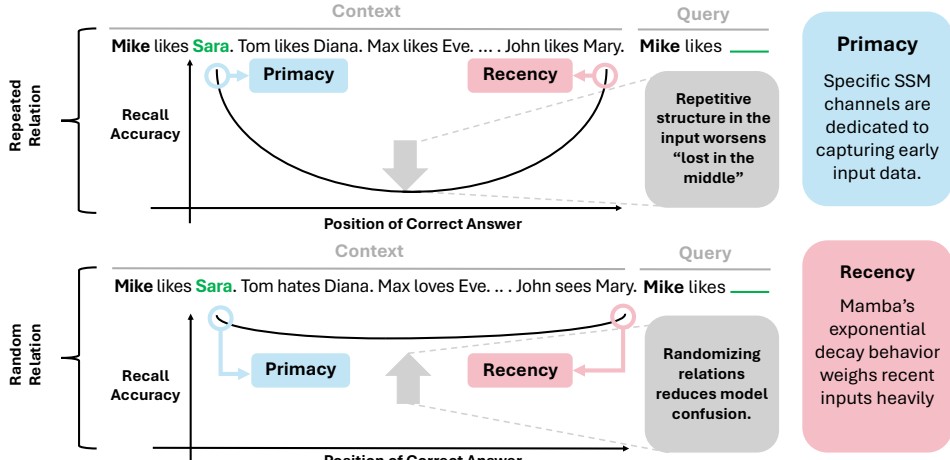

Figure 1: **Illustration of the structured recall task with two variants: repeated relation (top) and random relation (bottom).** The task measures Mamba's recall accuracy across sequence positions. **Repeated relation** reuses the same predicate (*likes*), producing strong **primacy** and **recency** effects, with a drop in middle recall. Primacy arises from early-tuned SSM channels; recency from Mamba's decay favors recent tokens. Repetition increases positional confusion, worsening the "lost-in-the-middle" effect. **Random relation** breaks this pattern, easing ambiguity and improving middle recall, though primacy and recency remain.

interaction with recency remains absent. To address this gap, we show that Mamba's dynamics naturally separate into two functional groups of channels: short-term channels that quickly decay, and a sparse subset of long-term channels that resists decay and carries early information forward. Furthermore, periodic inputs exacerbate forgetting (or decay) by shifting the effective dynamics toward larger discretization step size $\Delta_t$, thereby amplifying the decay effect. The distinction, however, between both channels arises from sensitivity to the step size $\Delta_t$. Although large $\Delta_t$ accelerates forgetting, the long-term channels remain stable by effectively maintaining smaller $\Delta_t$[1].

In this work we demonstrate how targeted channel ablations, state initialization, and periodic inputs modulate Mamba's temporal behavior, and set the stage for a unified understanding of position effects. Our main contributions are:

- Using structured recall tasks, we uncover a sparse subset of internal channels [2] in the selective state-space block of Mamba the persistently encode early tokens and support primacy effect (Section 4.2, Figure 3).

- Building on this discovery, we show that these primacy-related channels generalize beyond recall: interventions along them impair performance in both long-context question answering and summarization tasks (4.3).

- We demonstrate that repeated patterns exacerbate forgetting in Mamba, providing evidence of a strong lost-in-the-middle effect. Our analysis traces this to the interaction between repeated periodic inputs and the discretization mechanism, where the coupling of $\Delta_t$ with the $A$ and $B$ matrices drives a stronger tendency to forget. This mechanism underlies the "bottom part" of the U-shape observed in Mamba (Section 4.3, Figure 7).

- We demonstrate that introducing distractions markedly reduces recent-position recall accuracy, revealing how this behavior influences recency dynamics under interference (Section 4.4, Figure 5b).

- We systematically characterize a U-shaped accuracy profile across sequence positions in Mamba (Figure 1), demonstrating that state-space models, much like just transformers, exhibit joint primacy and recency effects.

---

[1]In Mamba, every channel has its own discretization step size $\Delta_t$.

[2]"Channels" refer to individual state dimensions within the selective state-space block, roughly analogous to embedding dimensions in a transformer.

## 2 RELATED WORK

Transformer-based language models have been shown to exhibit phenomena such as primacy and recency effects when evaluated on structured recall tasks (Janik, 2023). These effects emerge due to architectural asymmetries introduced by causal masking, learned positional embeddings, and attention dynamics (Wu et al., 2025). A key mechanism behind recall in transformers is the *induction head*, which allows the model to learn copy operations by attending to repeated patterns in the input (Olsson et al., 2022).

State space models (SSMs) process sequences through recursive updates to a fixed-size hidden state, enabling linear-time operation. Unlike transformers, they do not rely on token-to-token interactions and are therefore biased toward recent inputs, a recency effect formally analyzed in smoothing studies (Wang et al., 2025; Jelassi et al., 2024).

Primacy has also been observed in SSMs. Morita (2025) show that S4 models can exhibit strong primacy depending on how the discretization step $\mathbf{\Delta}_t$ is initialized. In time-invariant SSMs such as S4, $\mathbf{\Delta}_t$ defines the effective frequency spectrum the model can capture, a connection further explored from autocorrelation-based perspectives viewing SSMs as spectral filters (Liu & Li, 2024).

An illustrative case is S4D's diagonal initialization (Gu et al., 2022), where the continuous-time state matrix uses complex values $\Lambda_n = -\frac{1}{2} + 2\pi in$, which after discretization yields eigenvalues $\lambda_n = \exp(\mathbf{\Delta}_t \Lambda_n)$, setting the system's poles in the Z-domain. Pole magnitudes govern decay, while imaginary components define resonant frequencies. Thus $\mathbf{\Delta}_t$ directly controls frequency coverage and effective scale of information retention.

Unlike time-invariant models such as S4, Mamba (Gu & Dao, 2023) updates its state through input-dependent functions, including a learned discretization parameter $\mathbf{\Delta}_t$. This breaks many assumptions of a standard time-invariant setting. Although time-variant, $\mathbf{\Delta}_t$ still acts as a forgetting gate: smaller values extend information retention, while larger values accelerate decay and control input integration. Repeated or periodic inputs bias $\mathbf{\Delta}_t$ toward faster decay, an effect we explore in Section 4.3.

Recent work such as LongMamba (Ye et al., 2025) takes a performance-oriented view: it distinguishes local and global channels and applies a training-free filtering mechanism to enlarge Mamba's receptive field and improve long-context performance. In contrast, our study takes a mechanistic view, systematically characterizing the joint primacy and recency effects in Mamba. We show for the first time that Mamba exhibits a U-shaped recall profile and link these positional effects to identifiable architectural components.

## 3 MODEL ARCHITECTURE AND TASK

### 3.1 MAMBA ARCHITECTURE

Mamba is built on state space models (SSMs), which maintain structured hidden states $\mathbf{h}_t \in \mathbb{R}^{d \times N}$ that evolve over time through recurrent updates. Each input dimension $i \in \{1, \ldots, d\}$ of the embedding vector $\mathbf{x}_t \in \mathbb{R}^d$ is processed independently by a dedicated selective SSM block, enabling token-to-token interaction through the accumulation of information over time within the recurrent state dynamics. The state-update dynamics for each dimension are given by:

$$\mathbf{h}_t^{(i)} = \mathbf{A}_t^{(i)} \mathbf{h}_{t-1}^{(i)} + \mathbf{B}_t^{(i)} x_t^{(i)},$$
$$y_t^{(i)} = \mathbf{C}_t \mathbf{h}_t^{(i)} + D^{(i)} x_t^{(i)}, \tag{1}$$

where:

- $\mathbf{h}_t^{(i)} \in \mathbb{R}^N$ is the hidden state for dimesion $i$,

- $\mathbf{A}_t^{(i)} \in \mathbb{R}^{N \times N}$ is the recurrence matrix or the forget gate (diagonal in Mamba; possibly full or low-rank in models such as S4[3]),

---

[3]In S4D Gu et al. (2021), $\mathbf{A}$ is diagonal with entries initialized in the complex plane; $\mathbf{B}$ and $\mathbf{C}$ are also complex-valued, and are discretized via the bilinear transform. These variants differ in expressivity, stability, and frequency selectivity.

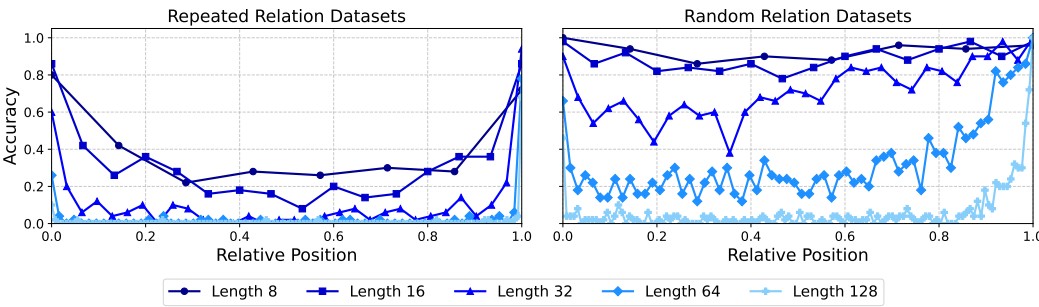

Figure 2: Recall accuracy of Falcon Mamba 7B as a function of position $k$ in the input sequence. Accuracy peaks at the beginning and end, forming a U-shaped curve characteristic of primacy and recency effects.

- $\mathbf{B}_t^{(i)} \in \mathbb{R}^N$, $\mathbf{C}_t \in \mathbb{R}^N$ are the input and output projection vectors (gates), respectively. $D^{(i)} \in \mathbb{R}$ is a residual weight,

- $x_t^{(i)} \in \mathbb{R}$ is the $i$-th dimension of the input at time $t$.

These models act as implicit convolutional filters in the frequency domain, managing long-range dependencies via kernel design rather than explicit attention. Mamba adds a learned per-token discretization parameter $\mathbf{\Delta}_t$ that dynamically modulates recurrence, adjusting how information is integrated or decayed with input structure. We conduct our experiments on *Falcon Mamba 7B* (Zuo et al., 2024) and validate on *Mamba 1.4B* (Gu & Dao, 2023) to ensure consistency across model sizes.

### 3.2 STRUCTURED RECALL TASK

To investigate how Mamba distributes and retains across time, we design a structured recall task inspired by serial position experiments in cognitive psychology (Murdock & Bennet, 1962). Each input sequence consists of $L$ subject–relation–object (`s-r-o`) triples, followed by a query targeting the object at a specific position in the sequence. The input fed to the model is structured as follows:

**Context**: `s₁ r₁ o₁. s₂ r₂ o₂. ⋯ s_L r_L o_L.` **Query**: `s_k r_k` (1≤k≤ L)

The model is expected to generate the correct `o_k` for the $k$-th triplet in the original sequence. To ensure consistent tokenization, we build the dataset from subject, relation, and object terms, each represented by a single unique token in the model's vocabulary, with all subjects and objects distinct within a sequence. For each target position $k$, we create 50 unique input sequences and report average accuracy to analyze recall performance across positions (details in Appendix A).

To probe position-related effects, we vary (i) **sequence length**, testing $L \in \{8, 16, 32, 64, 128\}$ triples to examine how recall changes with context size, and (ii) **relation type**, comparing **Repeated Relation** (all relation tokens identical, e.g., `Mike likes Sara. Tom likes Diana ... Mike likes`) versus **Random Relation** (relations uniquely sampled from a fixed vocabulary, introducing input diversity, e.g., `Mike likes Sara. Tom hates Diana ... Mike likes`).

## 4 EXPERIMENTAL RESULTS AND FINDINGS

### 4.1 CHARACTERIZING PRIMACY AND RECENCY EFFECTS IN MAMBA

Across sequence lengths and relation types, we consistently observe a U-shaped recall curve: higher accuracy at early and late positions with a dip in the middle (Figure 2), mirroring classic primacy and recency effects. The curve also depends on input structure: random relations yield better mid-position performance than repeated ones, whereas repetition produces a sharper U-shape with a deeper mid-position drop. This indicates that structural periodicity may cause the model to overweight positional edges, reinforcing primacy and recency. Having established these effects in Mamba's recall behavior, we next investigate how the primacy effect manifests internally.

## 4.2 PRIMACY: SPARSE SUBSET OF CHANNELS AND CAUSAL INTERVENTION

To understand how the primacy effect manifests within Mamba's internal components, we hypothesize that certain channels in the selective SSM blocks are responsible for persistently carrying information from early inputs across long contexts. These channels maintain their influence from early inputs over extended sequences.

Focusing on the recurrence from Equation 1, and assuming standard initialization $\mathbf{h}_0^{(i)} = 0$, the unrolled state dynamics yield:

$$\mathbf{h}_t^{(i)} = \sum_{j=1}^{t} \left( \prod_{k=j+1}^{t} \mathbf{A}_k^{(i)} \right) \mathbf{B}_j^{(i)} x_j^{(i)}. \tag{2}$$

We propose a quantitative test for the persistence of early-input contributions based on the magnitude of these terms. For each channel $i$, we compute a coefficient $\mathcal{M}^{(i)}$:

$$\mathcal{M}^{(i)} := diag(\prod_{t=2}^{T} \mathbf{A}_t^{(i)}) \in [0,1]^N \tag{3}$$

where $T$ is the final timestep in the context (prior to the query token). We can think of $\mathcal{M}^{(i)}$ as a measure of how strongly channel $i$ preserves the contribution of earlier inputs across the sequence. Here we focus only on the recurrent matrices $\mathbf{A}_t$, since $\mathbf{B}_t$ injects the current input at each step but does not accumulate over time (there is no $\mathbf{B}_{t-1}$ acting on $x_t$); thus it does not directly reflect the persistence of earlier inputs. We then define the long-term contribution probability of the channel $i$ given threshold $\tau \in [0,1]$ as:

$$P^{(i)}(\tau) := \frac{\left| \forall j \in [N], \ \mathcal{M}_j^{(i)} \geq \tau \right|}{N}, \tag{4}$$

where $[N] := \{1, 2, \ldots, N\}$. We define channel $i$ as long-term channel with probability $p \in [0,1]$ if $P^{(i)}(\tau) > p$. The complete identification pipeline is summarized in Algorithm 1 in the Appendix.

Using the procedure described above with threshold $\tau = 0.7$ and probability cut-off $p = 0.7$, we identify channels exhibiting long-term memory-like behavior. As shown in Figure 3a, these channels are not uniformly distributed but instead concentrate on specific layers of the Mamba architecture. In particular, for Falcon Mamba 7B, Layer 17 stands out with a disproportionately high number of long-term channels, suggesting a specialized role in preserving early input information.

**Causal intervention:** To test whether the identified channels truly preserve early inputs, we performed a controlled intervention on the forget gates $\mathbf{A}_t$, which scale past states in Mamba. We ablated gates of channels with high long-term contribution scores $P^{(i)}(0.7) > 0.7$, targeting the three layers with the highest density of such channels. This zeroed the recurrence matrices $\mathbf{A}_t$ at the first triplet, blocking early information from these long-term pathways.

We compared this targeted intervention with both a no-intervention baseline and a random ablation of the same number of channels. As shown in Figure 3b, only the targeted intervention significantly reduces first-position recall, whereas random ablation leaves performance largely intact. This confirms that early-input retention is concentrated in a structurally specialized subset of channels rather than uniformly distributed. We also experimented to zero the $\mathbf{A}_t$ for all time-steps in the identified channels as well as for doing intervention in Mamba 1.4B. The result can be seen in the Appendix D and E.

**Choice of $p$ and $\tau$:** To evaluate the sensitivity of our channel-identification method, we conducted an ablation study varying two key parameters: the threshold $\tau$, which controls how strict the criterion is for significant early-input retention, and $p$, the proportion of states exceeding this threshold. Notably, when $\tau = 1$, no states qualify. We experimented with combinations of $p \in \{0.5, 0.7, 0.9\}$ and $\tau \in \{0.5, 0.7, 0.9\}$, and report the impact of interventions targeting the top-1 layer identified under each criterion (Figure 4).

The results show a clear trend: high-precision selection of channels carrying early-input contributions (e.g., $p = 0.9$) with a moderately strict threshold ($\tau = 0.7$) is already sufficient to yield a

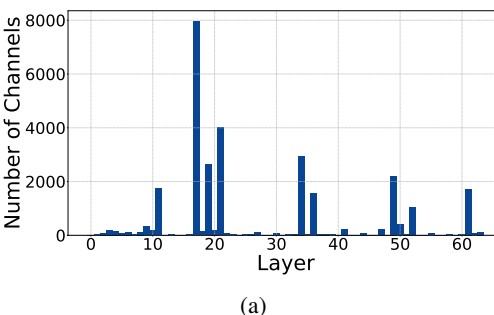 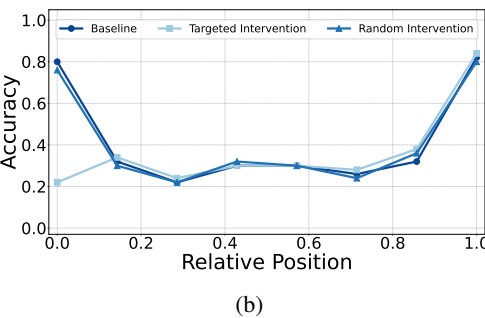

(a)

(b)

Figure 3: Layer-wise organization and functional role of long-term channels in Falcon Mamba 7B. **(a)** Distribution of long-term channels ($P^{(i)} > 0.7$ with $\tau = 0.7$) across Falcon Mamba 7B layers. **(b)** First-position recall drops sharply when intervening on the state matrix of identified long-term channels, confirming their role in retaining early input information.

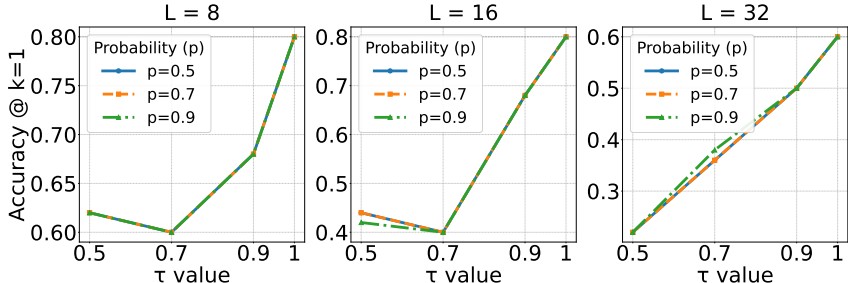

Figure 4: Impact of $\tau$ and $p$ on first-position recall under long-term channels interventions. High-precision selection (e.g., $p = 0.9$, $\tau = 0.7$) reliably degrades early recall, validating our identification method. The effect intensifies with longer sequences, while lower $\tau$ values recruit more channels.

significant drop in recall accuracy at the first position, demonstrating the effectiveness of our identification procedure. Additionally, as shown in the rightmost panel of Figure 4, the degradation in recall becomes more pronounced as the input sequence length increases, indicating that longer contexts rely more heavily on these identified channels. Finally, we observe that the optimal $\tau$ decreases with sequence length, suggesting that under relaxed selection criteria, a larger number of channels contribute to preserving information across long contexts.

**Initialization of state:** We tested an alternative recurrent-state initialization, replacing the default $\mathbf{h}_0^{(i)} = 0$ with values drawn from $\mathcal{U}(0, 1)$ one layer at a time. This introduces non-zero preactivations that can alter how early inputs are processed. Figure 5a shows that modifying Layer 31's initialization in the repeated-relation setting raises recall accuracy mainly at middle positions, mitigating the "lost in the middle" effect and flattening the U-shaped curve. This indicates that zero-initialized states may disproportionately favor early positions.

**Do these primacy-related channels have any effects on the downstream task?** Beyond the synthetic recall task, we tested whether the identified channels also support real-world performance by ablating them on long-context benchmarks, including research-paper QA (Qasper) (Dasigi et al., 2021), long-context understanding (MultiFieldQA-en) (Bai et al., 2024), multi-hop QA (2WikiMultihopQA) (Ho et al., 2020), and summarization (MultiNews) (Fabbri et al., 2019), all using the LongBench protocol (Bai et al., 2024). In this setting we zeroed all recurrence matrices $\mathbf{A}_t$ for the identified channels (rather than only three layers as in the recall task). As shown in Table 1, these interventions cause a substantially larger drop in long-context performance than random ablations, underscoring the channels' importance for handling information over long contexts.

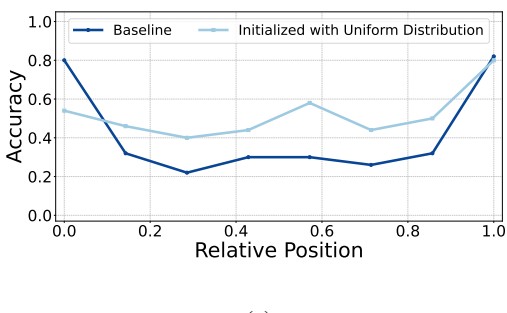 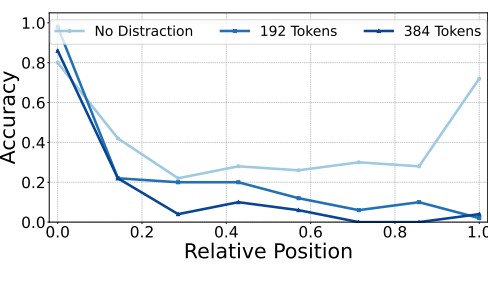

(a)           (b)

Figure 5: **(a)** Effect of initializing the recurrent state at Layer 31 using uniform values on repeated relation. **(b)** Adding distraction tokens disrupts the accuracy of the recall, especially in the recent positions.

Table 1: The effect of intervention on the performance of downstream tasks.

| Task | Metric | Type of Intervention | | |
|---|---|---|---|---|
| | | **Base** | **Long-term** | **Random** |
| Qasper | F1 | 27.88 | 20.68 | 23.94 |
| MultiFieldQA-en | F1 | 26.59 | 20.43 | 26.08 |
| 2WikiMultihopQA | F1 | 25.04 | 18.75 | 19.76 |
| MultiNews | Rouge-L | 25.21 | 20.5 | 24.84 |

### 4.3 INPUT PERIODICITY AND DELTA DYNAMICS

Moving from our structured recall task reveals an interesting observation regarding input periodicity. Sequences with repetitive structure (e.g. repeated relations) show stronger U-shaped recall compared to random sequences. We hypothesize that this is due to the behavior of Mamba's learned discretization factor $\mathbf{\Delta}_t$.

In Mamba, the per-token gate $\mathbf{\Delta}_t$ governs the model dynamics by controlling state updates. It is computed from the input $\mathbf{x}_t$ as

$$\mathbf{\Delta}_t = \text{softplus}(W_\Delta \mathbf{x}_t), \tag{5}$$

where $W_\Delta$ is a learned low-rank projection. This gate modulates both the recurrent transition and the input scaling:

$$\mathbf{A}_t = f_A(\mathbf{\Delta}_t), \tag{6}$$
$$\mathbf{B}_t = f_B(\mathbf{\Delta}_t) \tag{7}$$

so that large $\mathbf{\Delta}_t$ induces faster decay and stronger immediate input influence through $\mathbf{B}_t$, while small $\mathbf{\Delta}_t$ yields slower decay within $\mathbf{A}_t$ and greater influence of earlier inputs.

To test how $\Delta_t$ responds to input periodicity, we construct synthetic sequences of length 64 with controlled repetition frequencies. From a small vocabulary, we sample a token and repeat it every $k \in \{1, 4, 16, 64\}$ positions, where $k = 64$ corresponds to a fully random (non-repetitive) sequence. These inputs are passed to Mamba, and we record the $\Delta_t$ values across time and layers.

**Global Trends: Average $\Delta_t$ across Layers.** Figure 6a shows the delta values averaged across all layers and channels. We observe a clear monotonic relationship: sequences with longer repetition periods (i.e., lower frequency) result in higher average $\Delta_t$ values. This suggests that Mamba allocates more input responsiveness when the input varies slowly, while rapidly changing inputs (high-frequency repetition, e.g., $k = 1$) suppress $\Delta_t$, resulting in slower decay of past contributions.

**Layer-wise $\Delta_t$ Heatmaps.** To understand how this trend evolves across depth, we plot $\Delta_t$ values for each layer averaged over channels (Figure 7). In early layers, $\Delta_t$ shows temporal repetition aligned with the input frequency but without strong modulation in magnitude. As we go deeper,

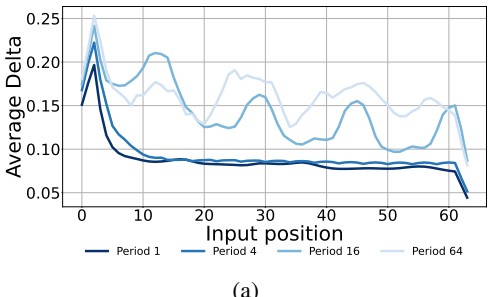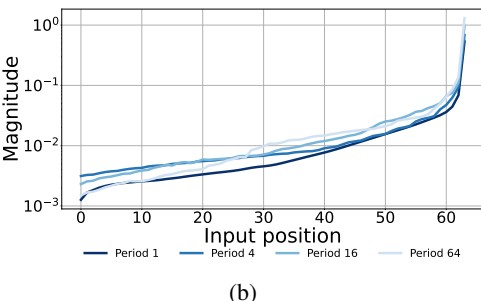

(a)                                                    (b)

Figure 6: **(a)** Average $\Delta$ across layers and channels for inputs with varying periodicities. Lower-frequency (longer-period) inputs induce larger $\Delta$ values, reflecting stronger integration of inputs. **(b)** Kernel magnitude across positions under different input periodicities. Lower-frequency (longer-period) inputs yield stronger kernel responses, especially for earlier positions, indicating greater past-time influence.

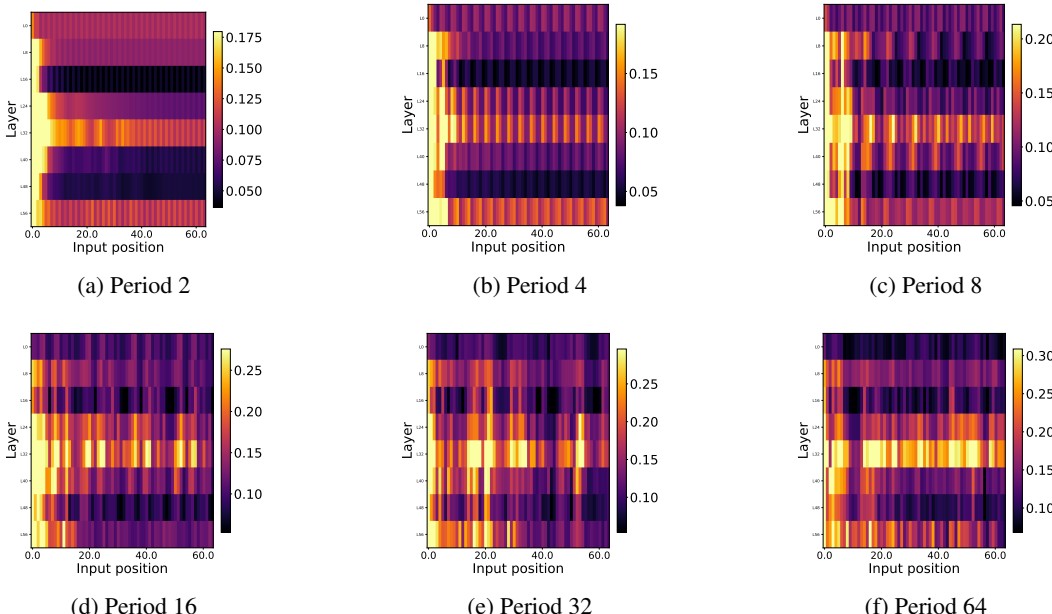

(a) Period 2                    (b) Period 4                    (c) Period 8

(d) Period 16                   (e) Period 32                   (f) Period 64

Figure 7: Average $\Delta$ per layer across input positions for periodic inputs. Early layers show strong correlation with input frequency: lower-frequency (longer-period) patterns yield smaller $\Delta$, indicating slower forgetting.

repetition in $\Delta_t$ values persists across time, reflecting continued sensitivity to periodic structure, and the scale of $\Delta_t$ increases, particularly for longer-period sequences. This indicates that deeper layers progressively reactivate input injection (via higher $\Delta_t$) when input frequency is low.

**Contributions to Final Token.** Finally, we compute the kernel (Equation 8) showing the contribution of each input position to the last token's hidden state (Figure 6b). We find that random sequences ($k = 64$) show strong contributions from early positions, whereas periodic sequences with small $k$ exhibit weaker contributions.

**Interpretation.** Together, these results reveal that Mamba's $\Delta_t$ gate adapts to input frequency in a non-trivial way: for high-frequency inputs it produces small $\Delta_t$ values, leading to slower decay of past contributions and weaker input injection, whereas for low-frequency inputs it produces large $\Delta_t$ values, resulting in faster decay and stronger input injection. This behavior could explain why repeated relations in the recall task lead to a stronger lost-in-the-middle effect: the model shifts from attending to immediate inputs to relying more on accumulated history.

## 4.4 RECENCY: EXPONENTIAL DECAY DYNAMICS

Finally, recency is the most consistent effect in our structured recall task: performance near the sequence end remains higher than in the middle even under input randomness (e.g., random relations). To explain this, we revisit exponential decay in Mamba's architectural design.

**Revisiting Exponential Decay Intuition.** Following the S6 analysis in Gu & Dao (2023), let the output at time $N$ be:

$$y_t^{(i)} = \mathbf{C}_t \sum_{j=1}^{t} \underbrace{\left( \prod_{k=j+1}^{t} \mathbf{A}_k^{(i)} \right) \mathbf{B}_j^{(i)} x_j^{(i)}}_{j^{th}-contribution}. \tag{8}$$

Clearly, the contribution of the $j^{th}$ inputs is proportional to $A^{t-j}$. In other words, inputs dominate exponentially with time. This is investigated as well in earlier studies Wang et al. (2025).

**The Effect of Distractors.** We empirically validate this dynamic by inserting random distractor tokens of varying lengths between the main sequence and the query prompt, as follow.

**Context**: $\mathtt{s_1}$ $\mathtt{r_1}$ $\mathtt{o_1.}$ $\mathtt{s_2}$ $\mathtt{r_2}$ $\mathtt{o_2.}\cdots$ $\mathtt{s_L}$ $\mathtt{r_L}$ $\mathtt{o_L.}$**Query**: $n-$random tokens. $\mathtt{s_k}$ $\mathtt{r_k}$ $\mathtt{(1\leq k\leq\ L)}$

As shown in Figure 5b, recall accuracy degrades consistently across all positions as the number of random tokens$n$ increases. This indicates that recency is not a fixed property of the architecture but is dynamically sustained by state transitions that gradually saturate over time; added distractors push the queried items deeper into the sequence, eroding their recency advantage and reinforcing the lost-in-the-middle effect.

## 5 DISCUSSIONS

Mamba consistently exhibits a U-shaped recall profile across input lengths and relation types. A small, non-uniform subset of channels drives primacy, while recency reflects its exponential-decay dynamics and disappears under distractors, showing it is sustained dynamically rather than hard-wired. Input periodicity and the discretization gate $\Delta_t$ jointly shape these effects: repeated patterns bias $\Delta_t$ toward faster decay and stronger input injection, amplifying the "lost-in-the-middle" effect, whereas altering state initialization can flatten the U-curve. Together, these results tie primacy, recency, and repetition effects to $\Delta_t$ and its interaction with a few long-range channels.

Although the recall task is synthetic, ablating these channels substantially degrades performance on long-context benchmarks, suggesting they also underpin real-world long-context reasoning. This indicates that position-dependent behaviors in state-space models arise from identifiable architectural features and highlights avenues for improvement, such as decoupling $\Delta_t$ from input injection, adding an auxiliary retention gate, or revisiting state-initialization strategies.

## 6 CONCLUSION

In this work we presented a unified, mechanistic analysis of position-dependent behavior in Mamba. Using structured recall tasks, we identified a sparse subset of channels within the selective state-space block that persistently encode early inputs and whose ablation markedly degrades long-context performance. We further showed that input periodicity biases the learned discretization gate, amplifying the "lost-in-the-middle" by reallocating information across positions, while recency emerges from Mamba's exponential-decay dynamics and collapses under distraction. Together, these findings establish that primacy, recency, and repetition effects in state-space models are not incidental but stem from identifiable architectural components.

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

# APPENDIX

## A  RECALL TASK CONSTRUCTION AND EXPERIMENTAL SETUP

**Task Construction:**  To evaluate position-related behavior in Mamba-based models, we designed a structured recall task consisting of subject–relation–object triplets. Each input sequence contains $L$ unique triplets followed by a single query in the same format (e.g., "Mike likes"), prompting the model to retrieve the correct object. The queries always target a specific position within the context (e.g., the first or last triplet), allowing us to probe primacy and recency effects.

We ensured that both subject and object tokens were proper names, while relation tokens were verbs commonly used in natural language. Across tasks, we varied the type of relation to prevent overfitting to a specific pattern. Examples include:

- **Likes:** Mike likes Todd. Henry likes Victor. . . . Mike likes, **answer:** *Todd*
- **Fights:** Austin fights Paula. Jake fights Noah. . . . Austin fights, **answer:** *Paula*
- **Affects:** Brian affects Isaac. Amy affects Ron. . . . Brian affects, **answer:** *Isaac*

Each triplet sequence is tokenized and passed to the model, with accuracy evaluated based on the model's top-1 prediction for the masked query position.

**Model Specifications:**  We evaluate two SSM-based models: **Mamba 1.4B**, the original open-weight Mamba model with 48 layers and state-space recurrence Gu & Dao (2023), and **Falcon Mamba 7B**, a larger and more scalable variant (Zuo et al., 2024).

Both models are evaluated in inference mode using greedy decoding. For analysis, all subject, relation, and object tokens are constrained to be single-token units. Recall is measured by the top-1 accuracy of the predicted object. Although sampling is non-deterministic, we generate 50 examples per target position to ensure robustness and mitigate randomness and lexical artifacts through averaging.

**Hardware Setup:** All experiments were run on a single NVIDIA RTX A6000 GPU with 48GB of memory.

---

**Algorithm 1** Identifying High-Contribution Dimensions Across Layers

---

**Require:** Input sequence, recurrence matrices $A_t$, threshold $\tau$, proportion $p$
  Initialize `layers_dim` as a dictionary for storing selected channels per layer
  **do** Perform a forward pass to collect $A_t$ and $B_t$ across all timesteps
  **for** each layer $\ell = 0$ to $L-1$ **do**
    `satisfying_dim` $\leftarrow []$
    **for** each channel $d = 0$ to $D-1$ **do**
      $A \leftarrow A_\ell[:, d, : t, :]$                               $\triangleright$ Recurrence across time
      $M \leftarrow \prod_{t=2}^{T} A_t$                       $\triangleright$ Element-wise product across time
      $\rho \leftarrow \text{mean}(\mathbb{I}[M > \tau])$
      **if** $\rho > p$ **then**
        Append $(d, \rho)$ to `satisfying_dim`
      **end if**
    **end for**
    `layers_dim[`$\ell$`]` $\leftarrow$ `satisfying_dim`
  **end for**

---

## B  ALGORITHM FOR IDENTIFYING LONG-TERM CHANNELS

We identify channels in Mamba that sustain early-input information across recurrent dynamics. The core hypothesis is that channels exhibiting strong cumulative recurrence from the first timestep carry long-term contributions.

Let $A_t^{(i)} \in \mathbb{R}^{N \times N}$ and $B_t^{(i)} \in \mathbb{R}^N$ denote the recurrence matrix and input projection at timestep $t$ for channel $i$, respectively. For each channel, we compute the cumulative product of $A_t^{(i)}$ across

time and determine the proportion of dimensions exceeding a recurrence threshold $\tau$. A channel is marked as a candidate if this proportion exceeds a threshold $p$.

Algorithm 1 summarizes the procedure. A forward pass on a single input sample is used to extract the relevant matrices. The output is a dictionary `layers_dim` mapping each layer to its identified channels, which are later used for targeted intervention (Section 4.2).

## C    ABLATION OF THE CHOICE OF $p$ AND $\tau$

We find that high-precision selection ($p = 0.9$) with a lenient threshold ($\tau = 0.5$) yields the greatest drop in recall. Increasing the number of intervened layers further amplifies this effect.

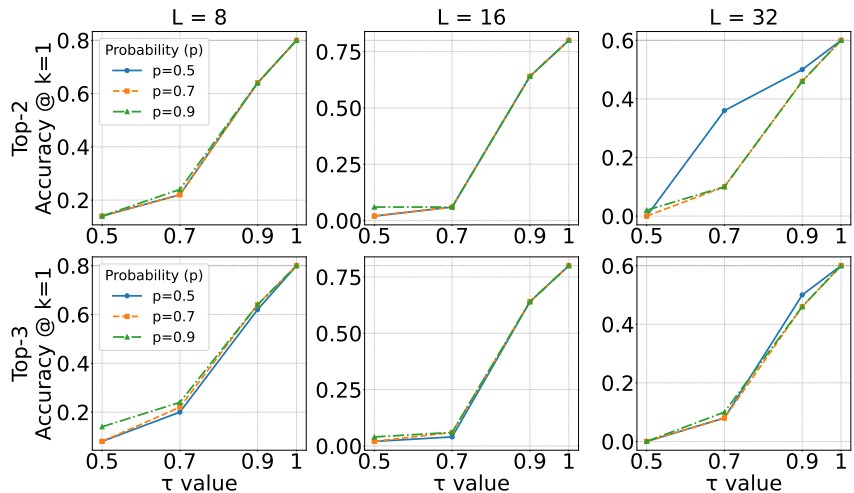

Figure 8: Ablation of the choice of $p$, $\tau$ at intervening at top-n layers

## D    INTERVENTION ON ALL TIMESTEPS AND LONGER SEQUENCES

To assess the impact of ablation across all time steps, we perform the intervention by turning off the recurrence matrices $\mathbf{A}_t$ for all timesteps. Figure 9 shows the resulting recall accuracy at each position. Targeted intervention produces a dramatic drop at the first position ($0.80 \rightarrow 0.28$)—far larger than at any other position. Random intervention, by contrast, causes only mild and inconsistent degradation. This highlights that the identified channels are especially critical for early-position recall.

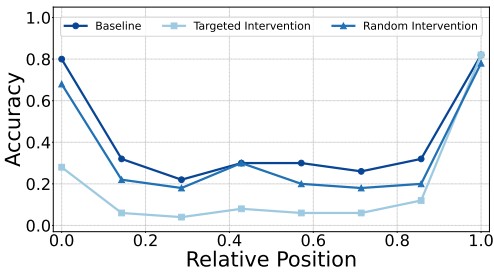

Figure 9: The result of ablating at all timestep on identified channels

We also evaluate the intervention method on longer sequences ($L = 16$, $L = 32$) using the same parameters ($p = 0.7$, $\tau = 0.7$). As shown in Figure 10, the intervention remains effective: recall at the first position drops substantially, indicating that the identified long-term channels continue to play a critical role as sequence length increases.

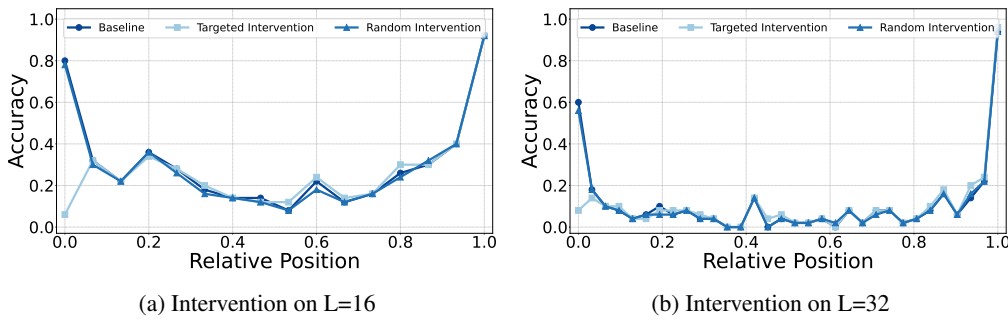

(a) Intervention on L=16

(b) Intervention on L=32

Figure 10: Intervention results for different sequence lengths.

# E   RESULTS ON DIFFERENT MODEL SIZE: MAMBA 1.4B

To test the generalization of our proposed method of localization and intervention, we conducted the same experiments in a different model variant, namely Mamba 1.4 B.

## E1   ABLATION OF THE CHOICE OF $p$ AND $\tau$

We begin by examining the behavior of different $p$ and $\tau$ configurations in Mamba 1.4B. The patterns observed in Falcon Mamba 7B hold consistently: even for shorter sequences, high-precision selection (e.g., $p = 0.9$, $\tau = 0.7$) already results in a noticeable drop in early recall. However, to achieve stronger intervention effects in Mamba 1.4B, it becomes necessary to increase the number of layers being ablated. This is likely due to the more distributed nature of long-term channels in smaller models. With fewer total parameters, the model may be forced to allocate long-term channels more sparsely across layers, requiring broader intervention to disrupt its recall capacity effectively.

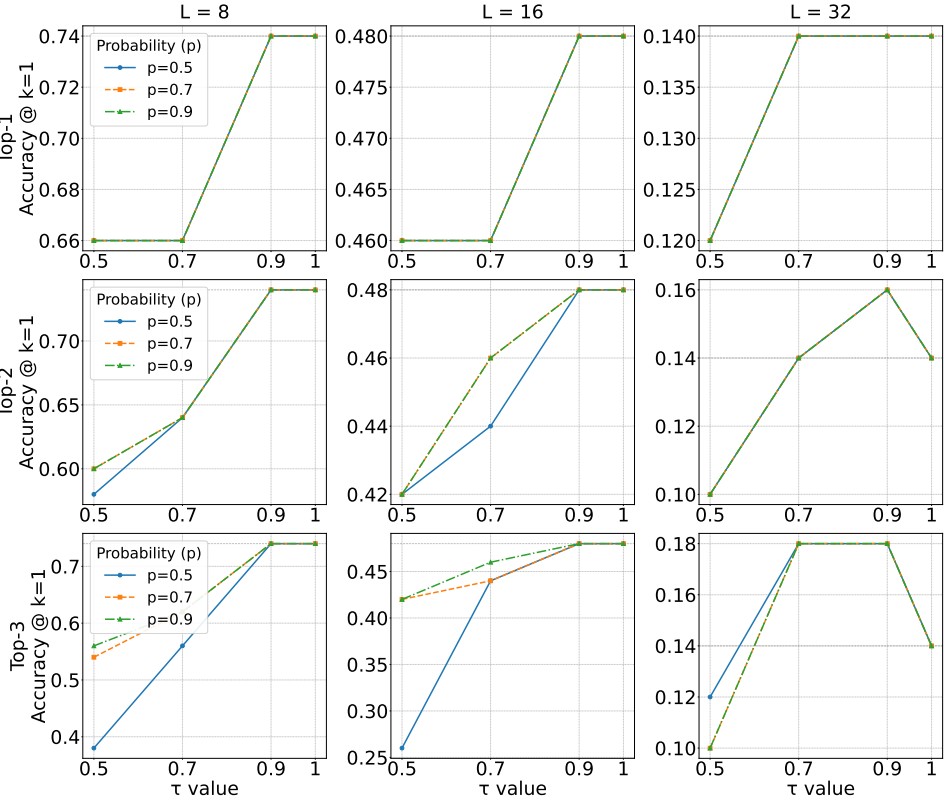

Figure 11: Ablation of the choice of $p$, $\tau$ at intervening at top-n layers for Mamba 1.4B

## E2    INTERVENTION RESULT ON VARIOUS LENGTH

We also apply the proposed intervention method to Mamba 1.4B. Based on the earlier ablation results, the largest drop in accuracy at $k = 1$ was observed using parameters $p = 0.5$, $\tau = 0.5$, and intervention on the top-3 layers. As shown in Figure 12, the intervention effectively impairs the model's ability to recall the object from the first triplet, demonstrating the generalizability of our identification method. However, for longer sequences (e.g., $L = 32$), the drop is less pronounced, likely because the model already struggles to retain early input in such settings—even without intervention.

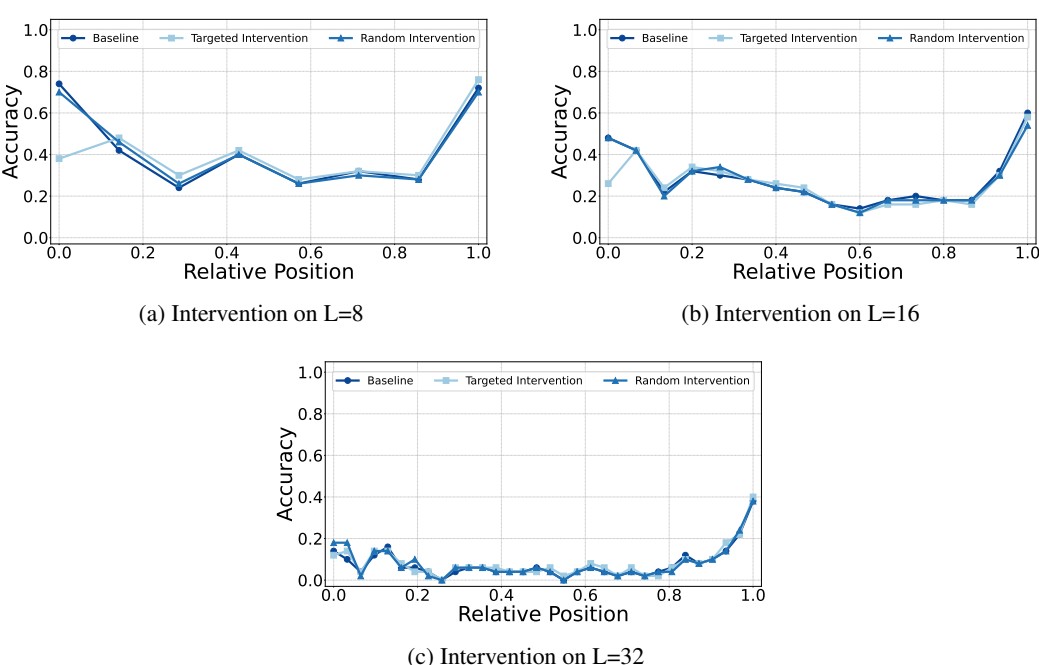

(a) Intervention on L=8

(b) Intervention on L=16

(c) Intervention on L=32

Figure 12: Intervention results for different sequence lengths on Mamba 1.4B.

## F    RESULTS ON MAMBA 2

We first investigate whether the primacy and recency effect persists in the latest Mamba variant, Mamba 2 (Dao & Gu, 2024). Using the same structured recall dataset, we observe that Mamba 2 exhibits a clear U-shaped recall curve, with elevated accuracy at both sequence boundaries, as shown by the baseline line in Figure 13b. This confirms that the primacy and recency effects are preserved even under the architectural modifications introduced in Mamba 2.

To examine the underlying mechanism, we adapt our analytical framework to the new model structure. Specifically, Mamba 2 operates at the *head level*, where all channels within a head share identical state-space dynamics. Accordingly, we redefine the channel-level coefficient in Eq. 3 at the head level. Each head $h$ maintains a scalar recurrence coefficient $\alpha_t^{(h)} \in [0, 1]$ that uniformly gates its internal state, corresponding to a recurrence matrix of the form $A_t^{(h)} = \alpha_t^{(h)}\mathbf{I}$. Since all elements of $A_t^{(h)}$ are identical, we remove the state dimension $N$ from the formulation and compute the coefficient as a scalar product across timesteps. We therefore define the head-level memory coefficient as

$$\mathcal{M}^{(h)} := \prod_{t=2}^{T} \alpha_t^{(h)}, \tag{9}$$

which quantifies the cumulative retention of early inputs across timesteps for head $h$. Using the same threshold criterion as in Section 4.2, $\tau = 0.7$, we classify heads satisfying $\mathcal{M}^{(h)} \geq \tau$ as *long-term heads*. Empirically, $9.32\%$ of all heads meet this condition, revealing a sparse subset

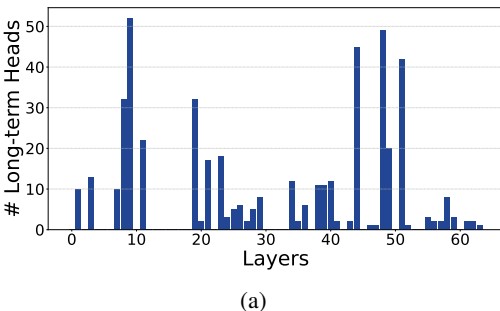 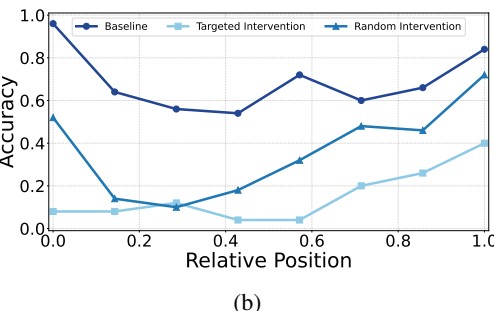

(a)                                           (b)

Figure 13: Layer-wise organization and functional role of long-term heads in Mamba 2. **(a)** Distribution of long-term heads ($\tau = 0.7$) across layers. **(b)** Targeted intervention on the state matrices of identified heads leads to a sharp drop in first-position recall and noticeable degradation at later positions, indicating that long-term heads are jointly responsible for early-information retention and context propagation.

that parallels the long-term channels identified in Mamba. Figure 13a shows that these long-term heads are non-uniformly distributed across layers, with dense clusters analogous to those observed in Falcon Mamba 7B (see Figure 3a).

Next, we perform an intervention similar to the Mamba 1 setting by zeroing the $A_t$ matrices for the identified heads at the first triplet. As shown in Figure 13b, the targeted intervention causes a pronounced drop in recall at the first position compared to the random baseline, confirming that these heads are crucial for preserving early-input information. Interestingly, the degradation extends beyond the first position. We attribute this to the coarser granularity of head-level manipulation in Mamba 2, where each head aggregates the dynamics of multiple channels that jointly contribute to the recurrent update. Disabling a single head, therefore, disrupts not only the pathways specialized for early-token retention but also potentially affects those supporting intermediate or context-compositional representations. In effect, the intervention introduces correlated perturbations across the state-space dimensions, weakening the recurrent propagation of information throughout the sequence. This results in a broader degradation pattern across positions, indicating that in Mamba 2, long-term heads participate in both early-information preservation and distributed context maintenance, rather than acting as purely localized carriers.

# G    INITIALIZATION ON DIFFERENT LAYERS

Here, we provide the complementary results of the initialization of the state, in which we initialize using values drawn from $\mathcal{U}(0, 1)$ one layer at a time.

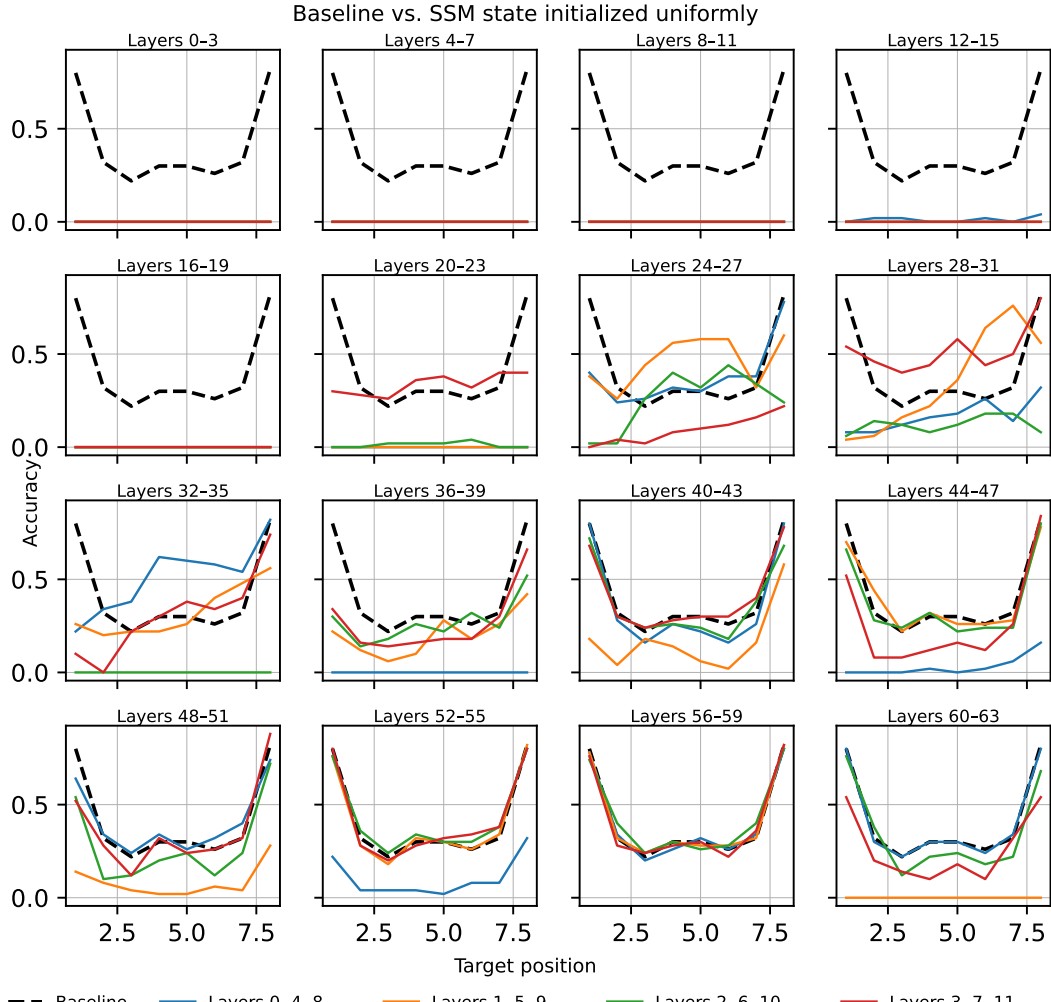

Figure 14: **Baseline vs. SSM state initialized uniformly.** Accuracy across target positions for the baseline model (dashed black) and SSM layers (colored solid lines). Each subplot groups four consecutive layers; colors repeat every four layers (blue = 0, 4, 8, . . . ; orange = 1, 5, 9, . . . ; green = 2, 6, 10, . . . ; red = 3, 7, 11, . . . ). Recall that accuracy changes when the hidden state is initialized using a non-zero vector.