# OpenReview forum: "Towards Understanding Primacy and Recency Effects in Mamba: A Mechanistic Perspective"
_ICLR.cc/2026/Conference — Submitted to ICLR 2026_

### Official Review · Reviewer_xxn1 · 2025-11-01

**Soundness:** 2
**Presentation:** 1
**Contribution:** 3
**Rating:** 4
**Confidence:** 3

**Summary:**

This paper identifies a subset of channels in Mamba models that are responsible for primacy and recency effects in recall tasks. These effects have previously been explored both behaviorally and mechanistically in Transformers, but are understudied in SSMs. They also show that recency effects "collapse" when the sequence model is shown distractor tokens, and that ablating the identified channels degrades performance on tasks "beyond" recall (e.g. long-context tasks).

**Strengths:**

* The mechanistic story is quite compelling, and I think that the experiments are well done
* I think it's really cool to see interp work done on non-Transformer models!

**Weaknesses:**

* The writing could use significant clarification; it took me a couple read throughs of the introduction to understand the main points (and I'm still not confident in my understanding). I also think that things should be restructured a bit, e.g. the experimental setup for 4.2/4.3/4.4 should be more clearly segmented from the results.
* I don't particularly know if these results are very significant. Primacy/recency effects are _interesting_, but it's not clear that developing a mechanistic understanding of these effects has any larger bearing on the field. I think the experiments that explore the effects of intervention on the recall-relevant channels are more interesting in this regard, but they aren't scoped out to the extent that I'd find compelling.

**Questions:**

See weaknesses above.

---

> ### Author Response · Authors · 2025-11-21
>
> >  **Reviewer xxn1 W1**: The writing could use significant clarification; it took me a couple read throughs of the introduction to understand the main points (and I'm still not confident in my understanding). I also think that things should be restructured a bit, e.g. the experimental setup for 4.2/4.3/4.4 should be more clearly segmented from the results.
>
> We thank the reviewer for this helpful feedback regarding clarity and structure. Our current organization was designed to parallel the three core phenomena studied, primacy, recency, and periodicity-induced effects, so that each section directly connects experimental observations with the underlying mechanism. However, we agree that the experimental setup could be more clearly delineated from the results. We will add a brief “Experimental Setup” subsection before each main results section (4.2–4.4) to improve readability and ensure that methodological details are clearly separated from the findings.
>
> > **Reviewer xxn1 W2**: I don't particularly know if these results are very significant. Primacy/recency effects are interesting, but it's not clear that developing a mechanistic understanding of these effects has any larger bearing on the field. I think the experiments that explore the effects of intervention on the recall-relevant channels are more interesting in this regard, but they aren't scoped out to the extent that I'd find compelling.
>
> In fact mechanistically motivated analyses have had a strong impact on the field, for instance, the discovery of induction heads grew from simple copy setups and now anchors our understanding of in-context learning in Transformers, shaping later methods and diagnostics [1]. However, we totally agree that primacy/recency effects are only useful if they bear on real model capability. Our claim is precisely that they do: the well-documented “lost-in-the-middle” failure mode in LLMs is a direct manifestation of primacy/recency that harms multi-document QA and long-context retrieval [2]. This is not a synthetic artifact but a practical limitation for real applications (assistants over long reports, RAG, multi-doc QA) and is precisely the profile we mechanistically explain and manipulate here. Finally the reviewer nails it that intervention experiments are most interesting when they transfer. In our paper, ablating the identified long-term channels (through recall synthetic tasks) degrades performance on downstream long-context tasks under LongBench (e.g., Qasper F1 27.9→20.7, MultiFieldQA-en 26.6→20.4, 2WikiMultihopQA 25→18.8, and MultiNews Rouge-L 25.2→20.5) while random ablations of the same size have a smaller effect. This supports that the channels we isolate are not task-specific artifacts but general long-term memory carriers in Mamba.
>
> [1] Olsson, C., Elhage, N., Nanda, N., Joseph, N., DasSarma, N., Henighan, T., ... & Olah, C. (2022). In-context learning and induction heads. arXiv preprint arXiv:2209.11895.
>
> [2] Liu, N. F., Lin, K., Hewitt, J., Paranjape, A., Bevilacqua, M., Petroni, F., & Liang, P. (2024). Lost in the middle: How language models use long contexts. Transactions of the Association for Computational Linguistics, 12, 157-173.

---

### Official Review · Reviewer_WrNK · 2025-11-03

**Soundness:** 2
**Presentation:** 3
**Contribution:** 2
**Rating:** 6
**Confidence:** 3

**Summary:**

This paper investigates position-dependent recall phenomena (primacy and recency effects) in Mamba, a state-space model architecture. The authors identify a sparse subset of channels responsible for early-input retention, demonstrate how input periodicity exacerbates "lost-in-the-middle" effects through the discretization mechanism, and show that recency emerges from exponential decay dynamics.

**Strengths:**

1. The long-term memory channels identification method is principled and is based on the cumulative recurrence matrix product.

2. Showing that identified channels impact real long-context benchmarks strengthens the claim that these are functionally important, not just artifacts of the synthetic task.

3. The paper is generally well-written with effective visualizations

**Weaknesses:**

1. Section 4.4 on recency largely reiterates known exponential decay dynamics from prior work (Wang et al., 2025). The distractor experiment adds empirical support but offers no new mechanistic insight. This section feels underdeveloped compared to the primacy analysis.

2. Mechanistic claims about $Δ_t$ and periodicity are underdeveloped.

**Questions:**

1. In Equation 3, you compute the product of diagonal entries. But $A^{(i)}_t ∈ R^{N×N}$ could have off-diagonal structure. How do you ensure you're only extracting diagonal recurrence?

2. For the initialization experiment (Figure 5a), why only Layer 31? Is this a long-term channel layer? What happens in other layers?

3. Table 1: The random ablation sometimes outperforms targeted ablation (e.g., 2WikiMultihopQA). This is counterintuitive—how do you explain this?

---

> ### Author Response · Authors · 2025-11-21
>
> > **Reviewer WrNK W1:** Section 4.4 on recency largely reiterates known exponential decay dynamics from prior work (Wang et al., 2025). The distractor experiment adds empirical support but offers no new mechanistic insight. This section feels underdeveloped compared to the primacy analysis.
>
> We appreciate the reviewer’s observation and agree that Section 4.4 largely revisits the known exponential-decay dynamics previously analyzed by Wang et al. (2025). We have made this scope explicit in the text. To add more, our intention was not to introduce a new form of recency, but rather to (i) quantify its magnitude within our structured recall framework, (ii) test whether distractors modulate this effect empirically, and (iii) provide a complete serial-position picture alongside the primacy analysis.
>
> > **Reviewer WrNK W2:** Mechanistic claims about $\Delta_t$ and periodicity are underdeveloped.
>
> We respectfully disagree with the reviewer’s assessment. Our motivation for analyzing the learned discretization parameter $\Delta_t$ stems from a clear empirical observation: Mamba’s recall accuracy drops markedly under repeated (periodic) relation inputs (Fig. 2). This systematic degradation prompted a closer examination of the underlying mechanism. In Mamba, $\Delta_t$  directly governs the trade-off between retaining past state information and integrating new input. Our analysis of $\Delta_t$  dynamics across layers and input frequencies (Fig. 6a) shows that periodic inputs consistently bias $\Delta_t$ toward smaller values. Since a smaller $\Delta_t$ corresponds to slower state updates and weaker input injection, this causes the model to rely more heavily on its previous hidden state rather than seeing new information from the input. We therefore conclude that the observed recall decline under periodic inputs arises naturally from $\Delta_t$ -driven state dynamics rather than from dataset artifacts.
>
> > **Reviewer WrNK Q1**: In Equation 3, you compute the product of diagonal entries. But could have off-diagonal structure. How do you ensure you're only extracting diagonal recurrence?
>
> We would like to clarify that the A matrices in Mamba are diagonal by design. Therefore, each channel operates independently along its own diagonal element. As a result, to compute the cumulative recurrence, we simply extract the diagonal entries of $A_t$ (as illustrated in equation 3)​ for each channel and multiply them across timesteps.
>
> > **Reviewer WrNK Q2**: For the initialization experiment (Figure 5a), why only Layer 31? Is this a long-term channel layer? What happens in other layers?
>
> We systematically tested the uniform initialization across all 64 layers in Falcon Mamba 7B and observed that initiating the state with a nonzero vector in early layers destroys the recall capabilities, some of the mid layers maintain the U-shape, and some of them increase the recall in the mid-position. Layer 31 was selected for presentation in Figure 5a because it clearly demonstrated the effect, specifically the flattening of the U-curve and improvement in middle-position recall. We provide the results for all layers in the newly appended Appendix G.
>
> > **Reviewer WrNK Q3**: Table 1: The random ablation sometimes outperforms targeted ablation (e.g., 2WikiMultihopQA). This is counterintuitive—how do you explain this?
>
> Both interventions degrade performance relative to the baseline, but the targeted long-term channel ablation causes a larger drop (6.29 points) compared to random ablation (5.28 points). The purpose of Table 1 is to show that the long-term channels identified through our recall-task methodology are not merely artifacts of the synthetic setup, but they play a functional role in real-world long-context understanding tasks.

---

### Official Review · Reviewer_PZgk · 2025-11-08

**Soundness:** 3
**Presentation:** 3
**Contribution:** 2
**Rating:** 4
**Confidence:** 4

**Summary:**

This paper focuses on the recency (high recall on recent tokens) and primacy (high recall on initial tokens) effects in SSMs. The main contribution is the finding that only a sparse subset of channels carries information from the first few tokens to the end of the sequence. The authors support this hypothesis with experiments on various datasets and models.

**Strengths:**

- The authors uncover a primacy mechanism in Mamba, where only a sparse subset of channels carries information from the first few tokens to the end of the sequence.
- They then design a probing experiment to validate this claim.

**Weaknesses:**

- The empirical validation is somewhat limited in scope, as the experiments are confined to two specific model instances (Falcon Mamba 7B and Mamba 1.4B).
- The paper's primary contribution appears to be incremental, as it largely builds upon existing frameworks. The central new discovery, while interesting, is presented without a substantial theoretical justification or a deep analysis of its underlying principles. The paper would be significantly strengthened by either providing a more rigorous theoretical grounding for this finding or by conducting a more in-depth empirical analysis.

Minor:
- Given that the paper's central focus is on the primacy effect, the authors might consider refining the title and abstract to more explicitly reflect this emphasis.
- The authors should consider exploring whether these effects influence overall model ability (e.g., performance on downstream tasks). If a significant impact is identified, the paper would be substantially strengthened by proposing or evaluating methods to mitigate these sequential biases.

**Questions:**

The original paper on recency finding [1] visualizes log-influential scores as part of its analysis. This paper, however, focuses solely on accuracy metrics. Could the authors elaborate on the decision to omit an analysis of log-influential scores?"

---

> ### Author Response · Authors · 2025-11-21
>
> > **Reviewer PZgK W1**: The empirical validation is somewhat limited in scope, as the experiments are confined to two specific model instances (Falcon Mamba 7B and Mamba 1.4B).
>
> We chose Falcon Mamba 7B as it represents the largest model in the Mamba family, and Mamba 1.4B as a mid-sized counterpart. Following the reviewer suggestion we now extended our results to the latest variant, Mamba-2 2.8B (including the presence of both primacy and recency effects on the same recall task). Our framework for identifying long-term channels in Mamba-1 can be naturally adapted to Mamba-2 by operating at the head level. In general, ablating these long-term heads by setting the corresponding A matrices to zero similarly degrades recall performance, particularly for early positions, and more than random ablations (results are added to appendix F). Below is the summary of the result for Mamba 2:
>
> **Per-position accuracy comparison across Baseline, Random, and Targeted settings in Mamba 2:**
>
> | **Position** | **Baseline** | **Random** | **Targeted** |
> |:-------------:|:------------:|:-----------:|:-------------:|
> | 1 | 0.960 | 0.520 | 0.080 |
> | 2 | 0.640 | 0.140 | 0.080 |
> | 3 | 0.560 | 0.100 | 0.120 |
> | 4 | 0.540 | 0.180 | 0.040 |
> | 5 | 0.720 | 0.320 | 0.040 |
> | 6 | 0.600 | 0.480 | 0.200 |
> | 7 | 0.660 | 0.460 | 0.260 |
> | 8 | 0.840 | 0.720 | 0.400 |
>
> > **Reviewer PZgK W2**: The paper's primary contribution appears to be incremental, as it largely builds upon existing frameworks. The central new discovery, while interesting, is presented without a substantial theoretical justification or a deep analysis of its underlying principles. The paper would be significantly strengthened by either providing a more rigorous theoretical grounding for this finding or by conducting a more in-depth empirical analysis.
>
> While our work builds upon existing state-space architectures, its primary contribution lies in providing the first mechanistic account of how primacy and recency effects arise in Mamba and how they interact through the learned discretization gate Δt. While prior studies have reported similar effects in different contexts, no work has linked them to identifiable architectural components mechanistaclly.
>
> In fact our analysis goes beyond descriptive observation: we (i) isolate a sparse subset of channels that preserve early inputs (Fig. 3), (ii) demonstrate their causal impact through targeted interventions that selectively impair early recall and long-context understanding (Table 1), and (iii) connect periodic input structure to Δt driven decay dynamics (Figs. 6–7). These findings offer a principled explanation of sequence-position effects in SSMs and establish a framework for mechanistic interpretability in SSM-based architectures.
>
> > **Reviewer PZgK W3**: Given that the paper's central focus is on the primacy effect, the authors might consider refining the title and abstract to more explicitly reflect this emphasis.
>
>  While the primacy mechanism is indeed a key finding, it emerges in interaction with recency and periodicty, both of which are essential to explaining the overall U-shaped recall profile. We therefore thought the current framing reflects the paper’s full scope.
>
> > **Reviewer PZgK W4**: The authors should consider exploring whether these effects influence overall model ability (e.g., performance on downstream tasks). If a significant impact is identified, the paper would be substantially strengthened by proposing or evaluating methods to mitigate these sequential biases.
>
> In fact the reviewer’s suggestion is already existed in Section 4.2 and Table 1: intervening on the identified primacy-related channels leads to substantial performance degradation across downstream long-context understanding benchmarks (Qasper, MultiFieldQA-en, 2WikiMultihopQA, and MultiNews).
>
> > **Reviewer PZgK Q1**: The original paper on recency finding [1] visualizes log-influential scores as part of its analysis. This paper, however, focuses solely on accuracy metrics. Could the authors elaborate on the decision to omit an analysis of log-influential scores?"
>
> While we did not explicitly use log-influence scores, we conducted a similar analysis in Figure 6(b) by examining the kernel magnitude across input positions. This reveals that contributions from recent tokens increase toward the end of the sequence, consistent with recency effects reported by Wang et al. (2025). Further we complemented this analysis with accuracy-based structured recall experiments, which capture the joint primacy–recency behavior more directly.

---

### Official Review · Reviewer_EWwn · 2025-11-08

**Soundness:** 2
**Presentation:** 3
**Contribution:** 2
**Rating:** 4
**Confidence:** 4

**Summary:**

This study investigates the primacy and recency effects (the "U-shaped recall curve") in SSMs. The authors focus on the primacy mechanism, identifying a sparse subset of internal channels that function as a long-term memory. They demonstrate that ablating these specific channels selectively degrades the model's recall of early information and impairs its overall performance on long-context tasks.

**Strengths:**

+ The authors test their sparse channel hypotheses under multiple conditions, strengthening their conclusions. This includes varying sequence lengths, comparing "Repeated Relation" inputs to "Random Relation" inputs, and using "distractor tokens" to test the fragility of the recency effect.

**Weaknesses:**

- The contribution can be somewhat limited. The paper's core novelty rests on identifying the "sparse channel hypothesis". However, the recency and primacy effects themselves are well-studied phenomena. The authors also acknowledge their observations are "common" (ln 24) and "investigated in early studies" (ln 447).

- Another central weakness is that the paper successfully demonstrates that this hypothesis holds but does not explore the reason or rationale why it exists. The contribution would be substantially stronger if the authors provided a theoretical grounding or an investigation into the architectural properties of the model that give rise to these observed sparse channels.

- While authors reveal the issues with Mamba, there is no solution provided. Do authors believe these problems are inherent and cannot be relieved easily?

**Questions:**

1. Have authors tried to reproduce the observation for more advanced SSM/linear attention architectures, e.g., Mamba2, DeltaNet?

---

> ### Author Response · Authors · 2025-11-21
>
> > **Reviewer EWwn W1**: The contribution can be somewhat limited. The paper's core novelty rests on identifying the "sparse channel hypothesis". However, the recency and primacy effects themselves are well-studied phenomena. The authors also acknowledge their observations are "common" (ln 24) and "investigated in early studies" (ln 447).
>
> Our core contribution is to present a unified mechanistic account linking primacy, recency, and periodicity within Mamba’s temporal dynamics. While Wang et al. (2025) previously analyzed recency, our work extends this by uncovering a sparse subset of long-term channels responsible for primacy, and by showing how these channels interact with Mamba’s discretization gate to produce the characteristic U-shaped recall profile. Thus, the novelty lies in connecting distinct positional effects under a single mechanistic framework, and further extending that to realistic tests on real model abilities by ablating the sparse channel identified from synthetics task on downstream long-context (i.e. transferability).
>
>
>
> > **Reviewer EWwn W2**: Another central weakness is that the paper successfully demonstrates that this hypothesis holds but does not explore the reason or rationale why it exists. The contribution would be substantially stronger if the authors provided a theoretical grounding or an investigation into the architectural properties of the model that give rise to these observed sparse channels.
>
> The focus in this work is primarily mechanistic, aiming to uncover how architectural components in Mamba give rise to position-dependent behaviors. We agree that a formal theoretical treatment would further enrich the contribution, and we view our work as a necessary empirical foundation toward that direction as the reviewer pointed out. Finally, we did discuss part of the rational in section 4.3 where we studied how $\Delta_t$ plays a role in strengthening the u-shape if the input is repeated.
>
> > **Reviewer EWwn W3**: While authors reveal the issues with Mamba, there is no solution provided. Do authors believe these problems are inherent and cannot be relieved easily?
>
> In fact we did not aim to give a solution and that requires another round of investigation (by proposing new architectures and testing them, which is beyond the current study). To add more:  our analysis naturally suggests potential architectural directions for future improvement. For instance, in section 4.3 we suggested decoupling the discretization gate from input injection (lines 470–471) to better balance information retention and input integration. We hope this clarifies our stance and highlights that our findings provide actionable insight for future model refinement.
>
> > **Reviewer EWwn Q1**: Have authors tried to reproduce the observation for more advanced SSM/linear attention architectures, e.g., Mamba2, DeltaNet?
>
> Following the reviewer's suggestion we include Mamba-2 2.8B to the list of tested models. The framework for identifying long-term channels in Mamba-1 is adapted to Mamba-2 by operating at the head level. In summary, ablating these long-term heads degrades recall performance for early positions more than random ablations (refer to the newly appended Appendix F for details). Below is the summary of the result of Mamba 2
>
> **Per-position accuracy comparison across Baseline, Random, and Targeted settings in Mamba 2:**
>
> | **Position** | **Baseline** | **Random** | **Targeted** |
> |:-------------:|:------------:|:-----------:|:-------------:|
> | 1 | 0.960 | 0.520 | 0.080 |
> | 2 | 0.640 | 0.140 | 0.080 |
> | 3 | 0.560 | 0.100 | 0.120 |
> | 4 | 0.540 | 0.180 | 0.040 |
> | 5 | 0.720 | 0.320 | 0.040 |
> | 6 | 0.600 | 0.480 | 0.200 |
> | 7 | 0.660 | 0.460 | 0.260 |
> | 8 | 0.840 | 0.720 | 0.400 |

---

### Author Response · Authors · 2025-12-03
**Final Comment to the AC**

We thank the reviewers and AC for their time. We briefly summarize how we addressed the main concerns.

Our main contribution is a mechanistic account of how primacy, recency, and periodicity jointly arise in Mamba via a sparse set of long-term channels and their interaction with the discretization gate. We show that ablating these channels, identified on a synthetic recall task, significantly degrades performance on real long-context benchmarks (e.g., Qasper, MultiFieldQA-en, 2WikiMultihopQA, and MultiNews Rouge-L), indicating that they are genuine long-term memory pathways rather than artifacts.


In response to concerns about generality, we extended our analysis to **Mamba-2 2.8B**, adapting our framework at the head level and showing that ablating identified long-term heads selectively harms early-token recall more than random ablations (Appendix F). We also added layer-wise results (Appendix G) and clarified the role of $\Delta_t$ under periodic inputs.

Given the strengthened empirical scope and clarified exposition, and in light of this special ICLR review process, we hope the reviewers have had the opportunity to reflect on our responses. In any case, we trust that the AC will carefully consider the additions and contributions of this work when making the final decision.

---

### Meta-Review · Area_Chair_NqfH · 2025-12-17

**Summary:**

This paper generated substantial discussion around the scope and depth of its mechanistic contribution, particularly whether the identification of sparse long-term channels in Mamba constitutes a sufficiently novel or generalizable insight beyond prior work on (sort of well-known) primacy/recency effects.

Reviewers appreciated the careful experimental design esp. the causal channel intervention, but repeatedly questioned whether the findings advance understanding beyond a refined empirical characterization of known phenomena. While the rebuttal clarified the authors’ framing and added supporting experiments, concerns remained about theoretical grounding, broader architectural implications, and the ultimate significance of the insights for model design or practice.

**Reviewer Concerns:**

I believe several concrete empirical concerns were partially addressed by the rebuttal, including extending experiments to Mamba-2, clarifying the ablation methodology, and strengthening evidence that the identified channels affect downstream long-context tasks.

However, the central concerns about incremental novelty, lack of a deeper theoretical or architectural explanation for why such sparse channels emerge, and the absence of concrete implications or remedies remain largely unresolved.

Reviewers also expressed reservations about the underdeveloped treatment of recency and periodicity relative to the primacy analysis, as well as the clarity and organization of the presentation. As a result, while the rebuttal improved confidence in the correctness of the experiments, it did not seem to revamp perceptions of the paper’s overall impact.

**Reviewer Scores:**

Based on the discussion, I would expect most reviewers’ scores to remain close to their original borderline assessments. In particular, reviewers who were initially lukewarm positive appear unlikely to strengthen their scores given the remaining concerns about novelty and significance. Overall, AC's educated guess is that the score distribution would likely stay clustered around the reject / marginal range rather than shifting above acceptance.

---

### Decision · Program_Chairs · 2026-01-26

Reject